# A Practical Algorithm for Distributed Clustering and Outlier Detection*

**Jiecao Chen**
Indiana University Bloomington
Bloomington, IN
jiecchen@indiana.edu

**Erfan Sadeqi Azer**
Indiana University Bloomington
Bloomington, IN
esadeqia@indiana.edu

**Qin Zhang**
Indiana University Bloomington
Bloomington, IN
qzhangcs@indiana.edu

## Abstract

We study the classic $k$-means/median clustering, which are fundamental problems in unsupervised learning, in the setting where data are partitioned across multiple sites, and where we are allowed to discard a small portion of the data by labeling them as outliers. We propose a simple approach based on constructing small summary for the original dataset. The proposed method is time and communication efficient, has good approximation guarantees, and can identify the global outliers effectively. To the best of our knowledge, this is the first practical algorithm with theoretical guarantees for distributed clustering with outliers. Our experiments on both real and synthetic data have demonstrated the clear superiority of our algorithm against all the baseline algorithms in almost all metrics.

## 1 Introduction

The rise of big data has brought the design of distributed learning algorithm to the forefront. For example, in many practical settings the large quantities of data are collected and stored at different locations, while we want to learn properties of the union of the data. For many machine learning tasks, in order to speed up the computation we need to partition the data into a number of machines for a joint computation. In a different dimension, since real-world data often contain background noise or extreme values, it is desirable for us to perform the computation on the "clean data" by discarding a small portion of the data from the input. Sometimes these outliers are interesting by themselves; for example, in the study of statistical data of a population, outliers may represent those people who deserve special attention. In this paper we study *clustering with outliers*, a fundamental problem in unsupervised learning, in the distributed model where data are partitioned across multiple sites, who need to communicate to arrive at a consensus on the cluster centers and labeling of outliers.

For many clustering applications it is common to model data objects as points in $\mathbb{R}^d$, and the similarity between two objects is represented as the Euclidean distance of the two corresponding points. In this paper we assume for simplicity that each point can be sent by *one unit* of communication. Note that when $d$ is large, we can apply standard dimension reduction tools (for example, the Johnson-Lindenstrauss lemma) before running our algorithms.

We focus on the two well-studied objective functions $(k,t)$-means and $(k,t)$-median, defined in Definition 1. It is worthwhile to mention that our algorithms also work for other metrics as long as the distance oracles are given.

**Definition 1 ($(k,t)$-means/median)** *Let $X$ be a set of points, and $k,t$ be two parameters. For the $(k,t)$-median problem we aim for computing a set of centers $C \subseteq \mathbb{R}^d$ of size at most $k$ and a set of outliers $O \subseteq X$ of size at most $t$ so that the objective function $\sum_{p \in X \setminus O} d(p,C)$ is minimized. For the $(k,t)$-means we simply replace the objective function with $\sum_{p \in X \setminus O} d^2(p,C)$.*

**Computation Model.** We study the clustering problems in the *coordinator model*, a well-adopted model for distributed learning Balcan *et al.* (2013); Chen *et al.* (2016); Guha *et al.* (2017); Diakonikolas *et al.* (2017). In this model we have $s$ sites and a central coordinator; each site can communicate with the coordinator. The input data points are partitioned among the $s$ sites, who, together with the coordinator, want to jointly compute some function on the global data. The data partition can be either adversarial or random. The former can model the case where the data points are independently collected at different locations, while the latter is common in the scenario where the system uses a dispatcher to randomly partition the incoming data stream into multiple workers/sites for a parallel processing (and then aggregates the information at a central server/coordinator).

In this paper we focus on the one-round communication model (also called the *simultaneous communication* model), where each site sends a sketch of its local dataset to the coordinator, and then the coordinator merges these sketches and extracts the answer. This model is arguably the most practical one since multi-round communication will cost a large system overhead.

Our goals for computing $(k,t)$-means/median in the coordinator model are the following: (1) to minimize the clustering objective functions; (2) to accurately identify the set of global outliers; and (3) to minimize the computation time and the communication cost of the system. We will elaborate on how to quantify the quality of outlier detection in Section 4.

**Our Contributions.** A natural way of performing distributed clustering in the simultaneous communication model is to use the two-level clustering framework (see e.g., Guha *et al.* (2003, 2017)). In this framework each site performs the first level clustering on its local dataset $X$, getting a subset $X' \subseteq X$ with each point being assigned a weight; we call $X'$ the *summary* of $X$. The site then sends $X'$ to the coordinator, and the coordinator performs the second level clustering on the union of the $s$ summaries. We note that the second level clustering is required to output at most $k$ centers and $t$ outliers, while the summary returned by the first level clustering can possibly have more than $(k+t)$ weighted points. The size of the summary will contribute to the communication cost as well as the running time of the second level clustering.

The main contribution of this paper is to propose a simple and practical summary construction at sites with the following properties.

1. It is extremely fast: runs in time $O(\max\{k, \log n\} \cdot n)$, where $n$ is the size of the dataset.
2. The summary has small size: $O(k \log n + t)$ for adversarial data partition and $O(k \log n + t/s)$ for random data partition.
3. When coupled with a second level (centralized) clustering algorithm that $\gamma$-approximates $(k,t)$-means/median, we obtain an $O(\gamma)$-approximation algorithm for distributed $(k,t)$-means/median.[2]
4. It can be used to effectively identify the global outliers.

We emphasize that both the first and the second properties are essential to make the distributed clustering algorithm scalable on large datasets. Our extensive set of experiments have demonstrated the clear superiority of our algorithm against all the baseline algorithms in almost *all* metrics.

To the best of our knowledge, this is the first practical algorithm with theoretical guarantees for distributed clustering with outliers.

**Related Work.** Clustering is a fundamental problem in computer science and has been studied for more than fifty years. A comprehensive review of the work on $k$-means/median is beyond the scope

of this paper, and we will focus on the literature for centralized/distributed $k$-means/median clustering *with* outliers and distributed $k$-means/median clustering.

In the centralized setting, several $O(1)$-approximation or $(O(1), O(1))$-approximation[3] algorithms have been proposed Charikar *et al.* (2001); Chen (2009). These algorithms make use of linear programming and need time at least $\Omega(n^3)$, which is prohibitive on large datasets. Feldman and Schulman (2012) studied $(k, t)$-median via *coresets*, but the running times of their algorithm includes a term $O(n(k + t)^{k+t})$ which is not practical.

Chawla and Gionis (2013) proposed for $(k, t)$-means an algorithm called $k$-means--, which is an iterative procedure and can be viewed as a generalization of Llyod's algorithm Lloyd (1982). Like Llyod's algorithm, the centers that $k$-means-- outputs are not the original input points; we thus cannot use it for the summary construction in the first level clustering at sites because some of the points in the summary will be the outliers we report at the end. However, we have found that $k$-means-- is a good choice for the second level clustering: it outputs exactly $k$ centers and $t$ outliers, and its clustering quality looks decent on datasets that we have tested, though it does not have any worst case theoretical guarantees.

Recently Gupta *et al.* (2017) proposed a local-search based $(O(1), O(k \log(n)))$-approximation algorithm for $(k, t)$-means. The running time of their algorithm is $\tilde{O}(k^2 n^2)$,[4] which is again not quite scalable. The authors mentioned that one can use the $k$-means++ algorithm Arthur and Vassilvitskii (2007) as a seeding step to boost the running time to $\tilde{O}(k^2(k + t)^2 + nt)$. We note that first, this running time is still worse than ours. And second, since in the first level clustering we only need a summary – all that we need is a set of weighted points that can be fed into the second level clustering at the coordinator, we can in fact directly use $k$-means++ with a budget of $O(k \log n + t)$ centers for constructing a summary. We will use this approach as a baseline algorithm in our experimental studies.

In the past few years there has been a growing interest in studying $k$-means/median clustering in the distributed models Ene *et al.* (2011); Bahmani *et al.* (2012); Balcan *et al.* (2013); Liang *et al.* (2014); Cohen *et al.* (2015); Chen *et al.* (2016). In the case of allowing outliers, Guha et al. Guha *et al.* (2017) gave a first theoretical study for distributed $(k, t)$-means/median. However, their algorithms need $\Theta(n^2)$ running time at sites and are thus again not quite practical on large-scale datasets. In a concurrent work, Li and Guo (2018) further reduced the value of the objective function, but the proposed method does not output the outliers.

We note that the $k$-means$\|$ algorithm proposed by Bahmani *et al.* (2012) can be extended (again by increasing the budget of centers from $k$ to $O(k \log n + t)$) and used as a baseline algorithm for comparison. The main issue with $k$-means$\|$ is that it needs $O(\log n)$ rounds of communication which holds back its overall performance.

## 2  The Summary Construction

In this section we present our summary construction for $(k, t)$-median/means in the centralized model. In Section 3 we will show how to use this summary construction for solving the problems in the distributed model. Table 1 is the list of notations we are going to use.

| $X$ | input dataset | $n$ | $n = |X|$, size of the dataset |
|---|---|---|---|
| $k$ | number of centers | $\kappa$ | $\kappa = \max\{k, \log n\}$ |
| $t$ | number of outliers | $O^*$ | outliers chosen by OPT |
| $\sigma$ | clustering mapping $\sigma : X \to X$ | $d(y, X)$ | $d(y, X) = \min_{x \in X} d(y, x)$ |
| $\phi_X(\sigma)$ | $\phi_X(\sigma) = \sum_{x \in X} d(x, \sigma(x))$ | $\phi(X, Y)$ | $\phi(X, Y) = \sum_{y \in Y} d(y, X)$ |
| $\mathsf{OPT}^{\mathrm{med}}_{k,t}(X)$ | $\min\limits_{\substack{O \subseteq X, |C| \leq k \\ |O| \leq t}} \sum\limits_{p \in X \backslash O} d(p, C)$ | $\mathsf{OPT}^{\mathrm{mea}}_{k,t}(X)$ | $\min\limits_{\substack{O \subseteq X, |C| \leq k \\ |O| \leq t}} \sum\limits_{p \in X \backslash O} d^2(p, C)$ |

Table 1: List of Notations

**Algorithm 1: Summary-Outliers**$(X, k, t)$

---

**Input** : dataset $X$, number of centers $k$, number of outliers $t$
**Output** : a weighted dataset $Q$ as a summary of $X$

1   $i \leftarrow 0, X_i \leftarrow X, Q \leftarrow \emptyset$
2   fix a $\beta$ such that $0.25 \leq \beta < 0.5$
3   $\kappa \leftarrow \max\{\log n, k\}$
4   let $\sigma : X \rightarrow X$ be a mapping to be constructed, and $\alpha$ be a constant to be determined in the analysis.
5   **while** $|X_i| > 8t$ **do**
6      construct a set $S_i$ of size $\alpha\kappa$ by random sampling (with replacement) from $X_i$
7      for each point in $X_i$, compute the distance to its nearest point in $S_i$
8      let $\rho_i$ be the smallest radius s.t. $|B(S_i, X_i, \rho_i)| \geq \beta|X_i|$. Let $C_i \leftarrow B(S_i, X_i, \rho_i)$
9      for each $x \in C_i$, choose the point $y \in S_i$ that minimizes $d(x, y)$ and assign $\sigma(x) \leftarrow y$
10     $X_{i+1} \leftarrow X_i \backslash C_i$
11     $i \leftarrow i + 1$
12   $r \leftarrow i$
13   for each $x \in X_r$, assign $\sigma(x) \leftarrow x$
14   for each $x \in X_r \cup (\cup_{i=0}^{r-1} S_i)$, assign weight $w_x \leftarrow |\sigma^{-1}(x)|$ and add $(x, w_x)$ into $Q$
15   **return** $Q$

---

## 2.1   The Algorithm

Our algorithm is presented in Algorithm 1. It works for both the $k$-means and $k$-median objective functions. We note that Algorithm 1 is partly inspired by the algorithm for clustering *without* outliers proposed in Mettu and Plaxton (2002). But since we have to handle outliers now, the design and analysis of our algorithm require new ideas.

For a set $S$ and a scalar value $\rho$, define $B(S, X, \rho) = \{x \in X \mid d(x, S) \leq \rho\}$. Algorithm 1 works in rounds indexed by $i$. Let $X_0 = X$ be the initial set of input points. The idea is to sample a set of points $S_i$ of size $\alpha k$ for a constant $\alpha$ (assuming $k \geq \log n$) from $X_i$, and grow a ball of radius $\rho_i$ centered at each $s \in S_i$. Let $C_i$ be the set of points in the union of these balls. The radius $\rho_i$ is chosen such that at least a constant fraction of points of $X_i$ are in $C_i$.

Define $X_{i+1} = X_i \backslash C_i$. In the $i$-th round, we add the $\alpha k$ points in $S_i$ to the set of centers, and assign points in $C_i$ to their nearest centers in $S_i$. We then recurse on the rest of the points $X_{i+1}$, and stop until the number of points left unclustered becomes at most $8t$. Let $r$ be the final value of $i$. Define the weight of each point $x$ in $\cup_{i=0}^{r-1} S_i$ to be the number of points in $X$ that are assigned to $x$, and the weight of each point in $X_r$ to be 1. Our summary $Q$ consists of points in $X_r \cup (\cup_{i=0}^{r-1} S_i)$ together with their weights.

## 2.2   The Analysis

We now try to analyze the performance of Algorithm 1. The analysis will be conducted for the $(k, t)$-median objective function, while the results also hold for $(k, t)$-means; we will discuss this briefly at the end of this section. Due to space constraints, all missing proofs in this section can be found in the supplementary material.

We start by introducing the following concept. Note that the summary constructed by Algorithm 1 is fully determined by the mapping function $\sigma$ ($\sigma$ is also constructed in Algorithm 1).

**Definition 2 (Information Loss)** *For a summary $Q$ constructed by Algorithm 1, we define the information loss of $Q$ as*

$$loss(Q) = \phi_X(\sigma).$$

*That is, the sum of distances of moving each point $x \in X$ to the corresponding center $\sigma(x)$ (we can view each outlier as a center itself).*

We will prove the following theorem, which says that the information loss of the summary $Q$ constructed by Algorithm 1 is bounded by the optimal $(k, t)$-median clustering cost on $X$.

**Theorem 1** *Algorithm 1 outputs a summary $Q$ such that with probability $(1 - 1/n^2)$ we have that $loss(Q) = O\left(\mathsf{OPT}_{k,t}^{\mathrm{med}}(X)\right)$. The running time of Algorithm 1 is bounded by $O(\max\{\log n, k\} \cdot n)$, and the size of the outputted summary $Q$ is bounded by $O(k \log n + t)$.*

The proof of this theorem relies on building an upper bound on $\phi_X(\sigma)$ and a lower bound on $\mathsf{OPT}_{k,t}^{\mathrm{med}}(X)$. Namely, $\phi_X(\sigma) = O(\sum_i \rho_i |D_i|)$ and $\mathsf{OPT}_{k,t}^{\mathrm{med}}(X) = \Omega(\sum_i \rho_i |D_i|)$, where $D_i = C_i \backslash O^*$, where $C_i$ is constructed in the $i$-th round of Algorithm 1 and $O^*$ is the set of outliers returned by the optimal algorithm. See the detailed proof in the supplementary material.

As a consequence of Theorem 1, we obtain by triangle inequality arguments the following corollary that directly characterizes the quality of the summary in the task of $(k, t)$-median. We include a proof in the supplementary material for completeness.

**Corollary 1** *If we run a $\gamma$-approximation algorithm for $(k, t)$-median on $Q$, we can obtain a set of centers $C$ and a set of outliers $O$ such that $\phi(X \backslash O, C) = O(\gamma \cdot \mathsf{OPT}_{k,t}^{\mathrm{med}}(X))$ with probability $(1 - 1/n^2)$.*

**The running time.** We now analyze the running time of Algorithm 1. At the $i$-th iteration, the sampling step at Line 6 can be done in $O(|X_i|)$ time. The nearest-center assignments at Line 7 and 9 can be done in $|S_i| \cdot |X_i| = O(\kappa |X_i|)$ time. Line 8 can be done by first sorting the distances in the increasing order and then scanning the shorted list until we get enough points. In this way the running time is bounded by $|X_i| \log |X_i| = O(\kappa |X_i|)$. Thus the total running time can be bounded by

$$\sum_{i=0,1,\ldots,r-1} O(\kappa |X_i|) = O(\kappa n) = O(\max\{\log n, k\} \cdot n),$$

where the first equation holds since the size of $X_i$ decreases geometrically, and the second equation is due to the definition of $\kappa$.

Finally, we comment that we can get a similar result for $(k, t)$-means by appropriately adjusting various constant parameters in the proof. Please refer to the supplementary material for a more detailed discussion.

## 2.3 An Augmentation

In the case when $t \gg k$, which is typically the case in practice since the number of centers $k$ does not scale with the size of the dataset while the number of outliers $t$ does, we add an augmentation procedure to Algorithm 1 to achieve a better practical performance. The pseudocode can be found in the supplementary materials and the full version of this paper.

The augmentation is as follows, after computing the set of outliers $X_r$ and the set of centers $S = \cup_{i=0}^{r-1} S_i$ in Algorithm 1, we sample randomly from $X \backslash (X_r \cup S)$ an additional set of center points $S'$ of size $|X_r| - |S|$. That is, we try to make the number of centers and the number of outliers in the summary to be balanced. We then reassign each point in the set $X \backslash X_r$ to its nearest center in $S \cup S'$. Denote the new mapping by $\pi$. Finally, we include points in $X_r$ and $S$, together with their weights, into the summary $Q$.

It is clear that the augmentation procedure preserves the size of the summary asymptotically. And by including more centers we have $loss(Q) \leq \phi_X(\pi) \leq \phi_X(\sigma)$, where $\sigma$ is the mapping returned by Algorithm 1. The running time will increase to $O(tn)$ due to the reassignment step, but our algorithm is still much faster than all the baseline algorithms, as we shall see in Section 4.

## 3 Distributed Clustering with Outliers

In this section we discuss distributed $(k, t)$-median/means using the summary constructed in Algorithm 1. Our main result is the following theorem, which is based on the work by Guha *et al.* (2003, 2017). The proof for this theorem can be found in the supplementary material.

**Theorem 2** *Suppose Algorithm 2 uses a $\gamma$-approximation algorithm for $(k, t)$-median in the second level clustering (Line 2). We have with probability $(1 - 1/n)$ that:*

---

**Algorithm 2: Distributed-Median**$(A_1, \ldots, A_s, k, t)$

---

**Input** : For each $i \in [s]$, Site $i$ gets input dataset $A_i$ where $(A_1, \ldots, A_s)$ is a random partition of $X$

**Output** : a $(k, t)$-median clustering for $X = \cup_{i \in [s]} A_i$

1  for each $i \in [s]$, Site $i$ constructs a summary $Q_i$ by running *Summary-Outliers*$(A_i, k, 2t/s)$ (Algorithm 1) and sends $Q_i$ to the coordinator

2  the coordinator then performs a second level clustering on $Q = Q_1 \cup Q_2 \cup \ldots \cup Q_s$ using an off-the-shelf $(k, t)$-median algorithm, and returns the resulting clustering.

---

- *it outputs a set of centers $C \subseteq \mathbb{R}^d$ and a set of outliers $O \subseteq X$ such that $\phi(X \backslash O, C) \leq O(\gamma) \cdot \mathsf{OPT}^{\mathrm{med}}_{k,t}(X)$;*

- *it uses one round of communication whose cost is bounded by $O(sk \log n + t)$;*

- *the running time at the $i$-th site is bounded by $O(\max\{\log n, k\} \cdot |A_i|)$, and the running time at the coordinator is that of the second level clustering.*

We note that in Mettu and Plaxton (2002) it was shown that under some mild assumption, $\Omega(kn)$ time is necessary for any $O(1)$-approximate randomized algorithm to compute $k$-median on $n$ points with nonnegligible success probability (e.g., $1/100$). Thus the running time of our algorithm is optimal up to a $\log n$ factor under the same assumption.

In the case that the dataset is adversarially partitioned, the total communication increases to $O(s(k \log n + t))$. This is because all of the $t$ outliers may go to the same site and thus $2t/s$ in Line 1 needs to be replaced by $t$.

Finally, we comment that the result above also holds for the summary constructed using the augumented version (Sec. 2.3), except, as discussed in Section 2, that the local running time at the $i$-th site will increase to $O(t|A_i|)$.

## 4  Experiments

### 4.1  Experimental Setup

#### 4.1.1  Datasets and Algorithms

Due to space constraints, we only present the experimental results for two data sets (`kddFull` and `kddSp`). One can find results for a number of other datasets in our supplementary materials and the full paper.

- `kddFull`. This dataset is from 1999 kddcup competition and contains instances describing connections of sequences of tcp packets. There are about $4.9$M data points. We only consider the $34$ numerical features of this dataset. We also normalize each feature so that it has zero mean and unit standard deviation. There are $23$ classes in this dataset, $98.3\%$ points of the dataset belong to 3 classes (`normal` $19.6\%$, `neptune` $21.6\%$, and `smurf` $56.8\%$). We consider small clusters as outliers and there are $45747$ outliers.

- `kddSp`. This data set contains about $10\%$ points of `kddFull` (released by the original provider). This dataset is also normalized and there are $8752$ outliers.

We comment that finding appropriate $k$ and $t$ values for the task of clustering with outliers is a separate problem, and is not part of the topic of this paper. In all our experiments, $k$ and $t$ are naturally suggested by the datasets we use.

We compare the performance of following algorithms, each of which is implemented using the MPI framework and run in the coordinator model. The data are randomly partitioned among the sites.

- `ball-grow`. Algorithm 2 proposed in this paper, with the augmented version Algorithm 1 for the summary construction. As mentioned we use $k$-`means--` as the second level clustering at Line 2. We fix $\alpha = 2$ and $\beta = 4.5$ in the subroutine Algorithm 1.

- `rand`. Each site constructs a summary by randomly sampling points from its local dataset. Each sampled point $p$ is assigned a weight equal to the number of points in the local dataset that are closer to $p$ than other points in the summary. The coordinator then collects all weighted samples from all sites and feeds to $k$-`means--` for a second level clustering.

- $k$-`means++`. Each site constructs a summary of the local dataset using the $k$-`means++` algorithm Arthur and Vassilvitskii (2007), and sends it to the coordinator. The coordinator feeds the unions all summaries to $k$-`means--` for a second level clustering.

- $k$-`means`$\|$. An MPI implementation of the $k$-`means`$\|$ algorithm proposed by Bahmani *et al.* (2012) for distributed $k$-means clustering. To adapt their algorithm to solve the outlier version, we increase the parameter $k$ in the algorithm to $O(k+t)$, and then feed the outputted centers to $k$-`means--` for a second level clustering.

### 4.1.2 Measurements

Let $C$ and $O$ be the sets of centers and outliers respectively returned by a tested algorithm. To evaluate the quality of the clustering results we use two metrics: (a) $\ell_1$-`loss` (for $(k,t)$-median): $\sum_{p \in X \setminus O} d(p, C)$; (b) $\ell_2$-`loss` (for $(k,t)$-means): $\sum_{p \in X \setminus O} d^2(p, C)$.

To measure the performance of outlier detection we use three metrics. Let $S$ be the set of points fed into the second level clustering $k$-`means--` in each algorithm, and let $O^*$ be the set of actual outliers (i.e., the ground truth), we use the following metrics: (a) `preRec`: the proportion of actual outliers that are included in the returned summary, defined as $\frac{|S \cap O^*|}{|O^*|}$; (b) `recall`: the proportion of actual outliers that are returned by $k$-`means--`, defined as $\frac{|O \cap O^*|}{|O^*|}$; (c) `prec`: the proportion of points in $O$ that are actually outliers, defined as $\frac{|O \cap O^*|}{|O|}$.

### 4.1.3 Computation Environments

All algorithms are implemented in C++ with Boost.MPI support. We use Armadillo Sanderson (2010) as the numerical linear library and -O3 flag is enabled when compile the code. All experiments are conducted in a PowerEdge R730 server equipped with 2 x Intel Xeon E5-2667 v3 3.2GHz. This server has 8-core/16-thread per CPU, 192GB Memeory and 1.6TB SSD.

## 4.2 Experimental Results

We now present our experimental results. All results take the average of 10 runs. In our supplementary material, results for more datasets can be found, but all the conclusions remain the same.

### 4.2.1 Quality

We first compare the qualities of the summaries returned by `ball-grow`, `rand` and $k$-`means`$\|$. Note that the size of the summary returned by `ball-grow` is determined by the parameters $k$ and $t$, and we can not control the exact size. In $k$-`means`$\|$, the summary size is determined by the sample ratio, and again we can not control the exact size. On the other hand, the summary sizes of `rand` and $k$-`means++` can be fully controlled. To be fair, we manually tune those parameters so that the sizes of summaries returned by different algorithms are roughly the same (the difference is less than 10%). In this set of experiments, each dataset is randomly partitioned into 20 sites.

Table 2 presents the experimental results on `kddSp` and `kddFull` datasets. We observe that `ball-grow` gives better $\ell_1$-`loss` and $\ell_2$-`loss` than $k$-`means`$\|$ and $k$-`means++`, and `rand` performs the worst among all.

For outlier detection, `rand` fails completely. In both `kddFull` and `kddSp`, `ball-grow` outperforms $k$-`means++` and $k$-`means`$\|$ in almost all metrics. $k$-`means`$\|$ slightly outperforms $k$-`means++`.

### 4.2.2 Communication Costs

We next compare the communication cost of different algorithms. Figure 1a presents the experimental results. The communication cost is measured by the number of points exchanged between the

| dataset | algo | summarySize | $\ell_1$-loss | $\ell_2$-loss | preRec | prec | recall |
|---|---|---|---|---|---|---|---|
| kddSp | ball-grow | 3.37e+4 | **8.00e+5** | **3.46e+6** | **0.6102** | **0.5586** | **0.5176** |
| | $k$-means++ | 3.37e+4 | 8.38e+5 | 4.95e+6 | 0.3660 | 0.3676 | 0.1787 |
| | $k$-means$\|$ | 3.30e+4 | 8.18e+5 | 4.19e+6 | 0.2921 | 0.3641 | 0.1552 |
| | rand | 3.37e+4 | 8.85e+5 | 1.06e+7 | 0.0698 | 0.5076 | 0.0374 |
| kddFull | ball-grow | 1.83e+5 | **7.38e+6** | **3.54e+7** | **0.7754** | **0.5992** | **0.5803** |
| | $k$-means++ | 1.83e+5 | 8.21e+6 | 4.65e+7 | 0.2188 | 0.2828 | 0.1439 |
| | $k$-means$\|$ | | does not stop after 8 hours | | | | |
| | rand | 1.83e+5 | 9.60e+6 | 1.11e+8 | 0.0378691 | 0.6115 | 0.0241 |

Table 2: Clustering quality. $k = 3$, $t = 8752$ for kddSp and $t = 45747$ for kddFull

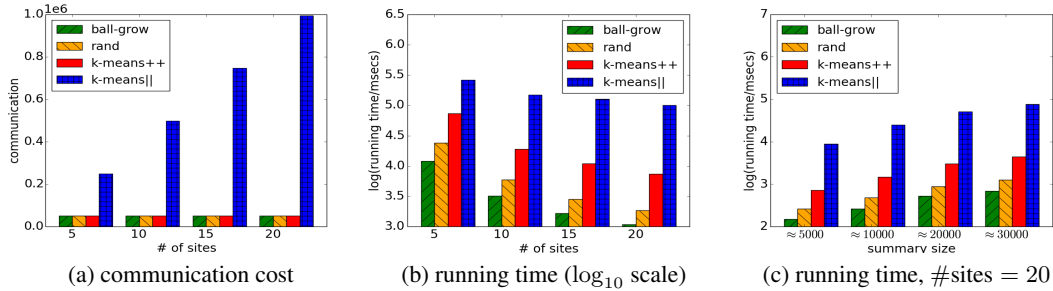

(a) communication cost      (b) running time ($\log_{10}$ scale)      (c) running time, #sites = 20

Figure 1: experiments on kddSp dataset

coordinator and all sites. In this set of experiments we only change the number of partitions (i.e., # of sites $s$). The summaries returned by all algorithms have almost the same size.

We observe that the communication costs of ball-grow, $k$-means++ and rand are almost independent of the number of sites. Indeed, ball-grow, $k$-means++ and rand all run in one round and their communication cost is simply the size of the union of the $s$ summaries. $k$-means$\|$ incurs significantly more communication, and it grows almost linearly to the number of sites. This is because $k$-means$\|$ grows its summary in multiple rounds; in each round, the coordinator needs to collect messages from all sites and broadcasts the union of those messages. When there are 20 sites, $k$-means$\|$ incurs 20 times more communication cost than its competitors.

### 4.2.3 Running Time

We finally compare the running time of different algorithms. All experiments in this part are conducted on kddSp dataset since $k$-means$\|$ does not scale to kddFull; similar results can also be observed on other datasets. The running time we show is only the time used to construct the input (i.e., the union of the $s$ summaries) for the second level clustering, and we do not include the running time of the second level clustering since it is always the same for all tested algorithms (i.e., the $k$-means--).

Figure 1b shows the running time when we change the number of sites while fix the size of the summary produced by each site. We observe that $k$-means$\|$ uses significantly more time than ball-grow, $k$-means++ and rand. This is predictable because $k$-means$\|$ runs in multiple rounds and communicates more than its competitors. ball-grow uses significantly less time than others, typically $1/25$ of $k$-means$\|$, $1/7$ of $k$-means++ and $1/2$ of rand. The reason that ball-grow is even faster than rand is that ball-grow only needs to compute weights for about half of the points in the constructed summary. As can be predicted, when we increase the number of sites, the total running time of each algorithm decreases.

We also investigate how the size of the summary will affect the running time. Note that for ball-grow the summary size is controlled by the parameter $t$. We fix $k = 3$ and vary $t$, resulting different summary sizes for ball-grow. For other algorithms, we tune the parameters so that they output summaries of similar sizes as ball-grow outputs. Figure 1c shows that when the size of summary increases, the running time increases almost linearly for all algorithms.

**Acknowledgments**

Jiecao Chen, Erfan Sadeqi Azer and Qin Zhang are supported in part by NSF CCF-1525024, NSF CCF-1844234 and IIS-1633215.

## Footnotes

*A full version of this paper is available at https://arxiv.org/abs/1805.09495

[2]We say an algorithm $\gamma$-approximates a problem if it outputs a solution that is at most $\gamma$ times the optimal solution.

[3]We say a solution is an $(a, b)$-approximation if the cost of the solution is $a \cdot C$ while excluding $b \cdot t$ points, where $C$ is the cost of the optimal solution excluding $t$ points.

[4]$\tilde{O}(\cdot)$ hides some logarithmic factors.

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
