[Supplementary Material]

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

## 2   Preliminaries

We are going to use the notations listed in Table 1.

We will also make use of the following lemmas.

**Lemma 1 (Chernoff Bound)** *Let $X_1, \ldots, X_n$ be independent Bernoulli random variables such that* $\mathbf{Pr}[X_i = 1] = p_i$. *Let $X = \sum_{i \in [n]} X_i$, and let $\mu = \mathbf{E}[X]$. It holds that $\mathbf{Pr}[X \geq (1 + \delta)\mu] \leq e^{-\delta^2 \mu/3}$ and $\mathbf{Pr}[X \leq (1 - \delta)\mu] \leq e^{-\delta^2 \mu/2}$ for any $\delta \in (0, 1)$.*

**Lemma 2 (Mettu and Plaxton (2002))** *Consider the classic balls and bins experiment where $b$ balls are thrown into $m$ bins, for some $b, m \in \mathbb{Z}^+$. Also, let $w_i$ be a weight associated with the $i$-th bin, for $i \in [m]$. Assuming, the probability of each ball falling into the $i$-th bin is $\frac{w_i}{\sum_{j=1}^{m} w_j}$ and $b \geq m$, the following holds:*

*For any $\epsilon \in \mathbb{R}^+$, there exists a $\gamma \in \mathbb{R}^+$ such that*

$$\mathbf{Pr}[\text{total weight of empty bins} > \epsilon \sum_i w_i] \leq e^{-\gamma b}.$$

*Note that the dependence of $\gamma$ on $\epsilon$ is independent of $b$ or $m$.*

| $X$ | input dataset | $n$ | $n = |X|$, size of the dataset |
|---|---|---|---|
| $k$ | number of centers | $\kappa$ | $\kappa = \max\{k, \log n\}$ |
| $t$ | number of outliers | $O^*$ | outliers chosen by OPT |
| $\sigma$ | clustering mapping $\sigma : X \to X$ | $d(y, X)$ | $d(y, X) = \min_{x \in X} d(y, x)$ |
| $\phi_X(\sigma)$ | $\phi_X(\sigma) = \sum_{x \in X} d(x, \sigma(x))$ | $\phi(X, Y)$ | $\phi(X, Y) = \sum_{y \in Y} d(y, X)$ |
| $B(S, X, \rho)$ | $= \{x \in X \mid d(x, S) \leq \rho\}$ | $r$ | # of iterations in Algo 1 |
| $X_i$ | remaining points at the $i$-th iteration of Algorithm 1 | $W_i$ | $X_i \backslash O^*$ |
| $C_i$ | clustered points at the $i$-th iteration of Algorithm 1 | $D_i$ | $C_i \backslash O^*$ |

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

*Proof:* Let $\pi : Q \to Q$ be the mapping returned by the $\gamma$-approximation algorithm for $(k, t)$-median on $Q$; we thus have $\pi(q) = q$ for all $q \in O$ and $\pi(X \backslash O) = C$. Let $\sigma : X \to X$ be the mapping returned by Algorithm 1 (i.e. $\sigma$ fully determines $Q$). We have that

$$
\begin{aligned}
\phi(X \backslash O, C) &\leq \sum_{x \in X} d(x, \pi(\sigma(x))) \\
&\leq \sum_{x \in X} \left( d(x, \sigma(x)) + d(\sigma(x), \pi(\sigma(x))) \right) \\
&= \sum_{x \in X} d(x, \sigma(x)) + \sum_{x \in X} d(\sigma(x), \pi(\sigma(x))) \\
&= loss(Q) + \sum_{q \in Q} w_q \cdot d(q, \pi(q)) \\
&= loss(Q) + \mathsf{SOL}_{k,t}^{\mathsf{med}}(Q),
\end{aligned}
$$

where $\mathsf{SOL}_{k,t}^{\mathsf{med}}(Q) = \sum_{q \in Q} w_q \cdot d(q, \pi(q))$ denotes the cost of the $\gamma$-approximation on $Q$. The corollary follows from Theorem 1 and Lemma 9 (set $s = 1$). $\square$

In the rest of this section we prove Theorem 1. We will start by bounding the information loss.

**Definition 3 ($O^*$, $W_i$ and $D_i$)** *Define $O^* \subseteq X$ to be the set of outliers chosen by running the optimal $(k, t)$-median algorithm on $X$; we thus have $|O^*| = t$. For $i = 0, 1, \ldots, r - 1$, define $W_i = X_i \backslash O^*$ and $D_i = C_i \backslash O^*$, where $X_i$ and $C_i$ are defined in Algorithm 1.*

We need the following utility lemma. It says that at each iteration in the while loop in Algorithm 1, we always make sure that at least half of the remaining points are not in $O^*$.

**Lemma 3** *For any $0 \leq i < r$, where $r$ is the total number of rounds in Algorithm 1, we have $2|W_i| \geq |X_i|$.*

*Proof:* According to the condition of the while loop in Algorithm 1 we have $|X_i| > 8t$ for any $0 \leq i < r$. Since $|O^*| = t$, we have

$$2|W_i| = 2|X_i \backslash O^*| \geq |X_i| + (|X_i| - 2|O^*|) \geq |X_i|.$$

$\square$

The rest of the proof for Theorem 1 proceeds as follows. We first show in Lemma 4 that $\mathrm{loss}(Q) = \phi_X(\sigma)$ can be upper bounded by $O(\sum_{0 \leq i < r} \rho_i |D_i|)$ (Lemma 4). We then show in Lemma 5 that $\mathsf{OPT}_{k,t}^{\mathrm{med}}(X)$ can be lower bounded by $\Omega(\sum_{0 \leq i < r} \rho_i |D_i|)$ with high probability (Lemma 5). Theorem 1 then follows.

**Lemma 4 (upper bound)** *It holds that*

$$\phi_X(\sigma) \leq 2 \sum_{0 \leq i < r} \rho_i |D_i|.$$

*Here $\rho_i$ is the radius we chosen in the $i$-th round of Algorithm 1.*

*Proof:* First, note that by Line 8 and the condition of the while loop in Algorithm 1 we have

$$|C_i| \geq \beta |X_i| \geq 8\beta t \overset{\beta \geq 0.25}{\geq} 2t. \tag{1}$$

We thus have by the definition of $D_i$ that

$$\begin{aligned}
|D_i| &= |C_i \backslash O^*| \geq |C_i| - |O^*| \\
&\overset{|O^*|=t}{=} |C_i| - t \\
&\overset{\text{by (1)}}{\geq} |C_i|/2. 
\end{aligned} \tag{2}$$

Observe that $X \backslash X_r = \cup_{0 \leq i < r} C_i$ and $C_i \cap C_j = \emptyset$ for any $i \neq j$, we can bound $\phi_{X \backslash X_r}(\sigma)$ by the following.

$$\begin{aligned}
\phi_{X \backslash X_r}(\sigma) &= \sum_{0 \leq i < r} \phi_{C_i}(\sigma) \\
&\leq \sum_{0 \leq i < r} \rho_i |C_i| \\
&\overset{\text{by (2)}}{\leq} \sum_{0 \leq i < r} 2\rho_i |D_i|.
\end{aligned}$$

The lemma follows since by our construction at Line 13 we have $\phi_{X_r}(\sigma) = 0$. $\square$

We now turn to the lower bound of $\mathsf{OPT}_{k,t}^{\mathrm{med}}(X)$.

**Lemma 5 (lower bound)** *It holds that*

$$\mathsf{OPT}_{k,t}^{\mathrm{med}}(X) = \Omega\left( \sum_{0 \leq i < r} \rho_i |D_i| \right).$$

Before proving the lemma, we would like to introduce a few more notations.

**Definition 4 ($\rho_i^{\mathrm{opt}}$ and $h$)** *Let $h = \frac{1+2\beta}{2}$; we thus have $1 > h > 2\beta > 0$ (recall in Algorithm 1 that $\beta < 0.5$ is a fixed constant). For any $0 \leq i < r$, let $\rho_i^{\mathrm{opt}} > 0$ be the minimum radius such that there exists a set $Y \subseteq X \backslash O^*$ of size $k$ with*

$$|B(Y, W_i, \rho_i^{\mathrm{opt}})| \geq h|W_i|. \tag{3}$$

The purpose of introducing $\rho_i^{\text{opt}}$ is to use it as a bridge to connect $\text{OPT}_{k,t}^{\text{med}}(X)$ and $\rho_i$. We first have the following.

**Lemma 6** $\text{OPT}_{k,t}^{\text{med}}(X) = \Omega\left(\sum_{0 \le i < r} \rho_i^{\text{opt}} |D_i|\right).$

Fix an arbitrary set $Y \subseteq X \backslash O^*$ of size $k$ as centers. To prove Lemma 6 we will use a charging argument to connect $\text{OPT}_{k,t}^{\text{med}}(X)$ and $\sum_{0 \le i < r} \rho_i^{\text{opt}} |D_i|$. To this end we introduce the following definitions and facts.

**Definition 5** ($E_i$, $E_i^m$ **and** $\mathcal{P}_\ell^m$) *For each $0 \le i < r$, define $E_i = \{x \in W_i \mid d(x, Y) \ge \rho_i^{\text{opt}}\}$. For any $m \in \mathbb{Z}^+$, define $E_i^m = E_i \backslash (\cup_{j>0} E_{i+jm})$. Let $\mathcal{P}_\ell^m = \{0 \le i < r \mid i \equiv \ell \pmod{m}\}$.*

Clearly, if $i \ne j$ and $j \equiv i \pmod{m}$, then $E_i^m$ and $E_j^m$ are disjoint. This leads to the following fact.

**Fact 1** *For any $i = 0, 1, \ldots, r-1$, we have*

$$\phi(Y, \cup_{i \in \mathcal{P}_\ell^m} E_i^m) = \sum_{i \in \mathcal{P}_\ell^m} \phi(Y, E_i^m)$$
$$\ge \sum_{i \in \mathcal{P}_\ell^m} \rho_i^{\text{opt}} |E_i^m|.$$

By the definitions of $\rho_i^{\text{opt}}$ and $E_i$ we directly have:

**Fact 2** *For any $i = 0, 1, \ldots, r-1$, $|E_i| \ge (1-h)|W_i|$.*

Let $z = \lceil \log_{1-\beta} \frac{1-h}{6} \rceil$ (a constant), we have

**Fact 3** *For any $i = 0, 1, \ldots, r-1$, $|E_i^z| \ge |E_i|/2$.*

*Proof:* We first show that $|E_i|, |E_{i+z}|, \ldots$ is a geometrically decreasing sequence.

$$
\begin{aligned}
|E_{i+z}| &\le |X_{i+z}| \\
&\le (1-\beta)^z |X_i| \\
&\overset{\text{Lemma 3}}{\le} 2(1-\beta)^z |W_i| \\
&\overset{\text{Fact 2}}{\le} \frac{2(1-\beta)^z}{1-h} |E_i| \\
&\overset{\text{Def. of } z}{\le} \frac{|E_i|}{3}.
\end{aligned}
$$

As a result, we have that $E_i^z$ holds a least a constant fraction of points in $E_i$.

$$
\begin{aligned}
|E_i^z| &= |E_i \backslash \cup_{j>0} E_{i+jz}| \\
&\ge |E_i| - \sum_{j>0} \frac{|E_i|}{3^j} \\
&\ge \frac{|E_i|}{2}.
\end{aligned}
$$

$\square$

**Fact 4** *For any $i = 0, 1, \ldots, r-1$, $|E_i^z| \ge (1-h)|D_i|/2$.*

*Proof:*

$$|E_i^z| \overset{\textbf{Fact 3}}{\ge} |E_i|/2 \overset{\textbf{Fact 2}}{\ge} (1-h)|W_i|/2 \overset{D_i \subseteq W_i}{\ge} (1-h)|D_i|/2.$$

$\square$

*Proof:*(of Lemma 6) Let
$$\ell = \operatorname{argmax}_{0 \leq j < z} \left( \textstyle\sum_{i \in \mathcal{P}_j^z} |E_i^z| \right).$$
Then $\phi(Y, X \backslash O^*)$ is at least

$$
\begin{aligned}
\phi(Y, \cup_{i \in \mathcal{P}_\ell^z} E_i^z) &\overset{\textbf{Fact } 1}{\geq} \sum_{i \in \mathcal{P}_\ell^z} \rho_i^{\mathtt{opt}} |E_i^z| \\
&\overset{\text{def. of } \ell}{\geq} \frac{1}{z} \sum_{0 \leq i < r} \rho_i^{\mathtt{opt}} |E_i^z| \\
&\overset{\textbf{Fact } 4}{\geq} \Omega(1) \cdot \sum_{0 \leq i < r} \rho_i^{\mathtt{opt}} |D_i|.
\end{aligned}
$$

The lemma then follows from the fact that $Y$ is chosen arbitrarily. □

Note that Lemma 6 is slightly different from Lemma 5 which is what we need, but we can link them by proving the following lemma.

**Lemma 7** *With probability $1 - 1/n^2$, we have $\rho_i^{\mathtt{opt}} \geq \rho_i/2$ for all $0 \leq i < r$.*

*Proof:* Fix an $i$, and let $Y \subseteq X \backslash O^*$ be a set of size $k$ such that $|B(Y, W_i, \rho_i^{\mathtt{opt}})| \geq h|W_i|$. Let $G = B(Y, W_i, \rho_i^{\mathtt{opt}})$. We assign each point in $G$ to its closest point in $Y$, breaking ties arbitrarily. Let $P_x$ be the set of all points in $G$ that are assigned to $x$; thus $\{P_x \mid x \in Y\}$ forms a partition of $G$.

Recall that $S_i$ in Algorithm 1 is constructed by a random sampling. Define
$$G' = \{y \in G \mid \exists x \in Y \ s.t. \ (y \in P_x) \wedge (S_i \cap P_x \neq \emptyset)\}.$$
We have the following claim.

**Claim 1** *For any positive constant $\epsilon$, there exists a sufficiently large constant $\alpha$ (Line 4 in Algorithm 1) such that*
$$|G'| \geq (1 - \epsilon)|G| \tag{4}$$
*with probability $1 - 1/n^2$.*

Note that once we have (4), we have that for a sufficiently small constant $\epsilon$,

$$
\begin{aligned}
|G'| &\geq (1 - \epsilon)|G| \\
&\overset{\text{Def. } 4}{\geq} (1 - \epsilon)h|W_i| \\
&\overset{h > 2\beta}{\geq} 2\beta|W_i| \\
&\overset{\text{Lemma } 3}{\geq} \beta|X_i|.
\end{aligned}
$$

Since $G' \subseteq B(S_i, W_i, 2\rho_i^{\mathtt{opt}}) \subseteq B(S_i, X_i, 2\rho_i^{\mathtt{opt}})$, we have $|B(S_i, X_i, 2\rho_i^{\mathtt{opt}})| \geq \beta|X_i|$. By the definition of $\rho_i$, we have that $\rho_i \leq 2\rho_i^{\mathtt{opt}}$. The success probability $1 - 1/n$ in Lemma 7 is obtained by applying a union bound over all $O(\log n)$ iterations.

Finally we prove Claim 1. By the definition of $G$ and Lemma 3 we have
$$|G| \geq h|W_i| \geq h/2 \cdot |X_i|. \tag{5}$$
Denote $S_i = \{s_1, \ldots, s_{\alpha\kappa}\}$. Since $S_i$ is a random sample of $X_i$ (of size $\alpha\kappa$), by (5) we have that for each point $j \in [\alpha\kappa]$, $\mathbf{Pr}[s_j \in G] \geq h/2$. For each $j \in [\alpha\kappa]$, define a random variable $Y_j$ such that $Y_j = 1$ if $s_j \in G$, and $Y_j = 0$ otherwise. Let $Y = \sum_{i \in [\alpha\kappa]} Y_j$; we thus have $\mathbf{E}[Y] \geq h/2 \cdot \alpha\kappa$. By applying Lemma 1 (Chernoff bound) on $Y_j$'s, we have that for any positive constant $\gamma$ and $h$, there exists a sufficiently large constant $\alpha$ (say, $\alpha = 10h/\gamma^2$) such that

$$
\begin{aligned}
\mathbf{Pr}[Y \geq \gamma\kappa] &\geq 1 - e^{-\left(\frac{2\gamma}{h}\right)^2 \cdot \frac{h}{2}\alpha\kappa/2} \\
&\geq 1 - 1/n^3,
\end{aligned}
$$

In other words, with probability at least $1 - 1/n^3$, $|S_i \cap G| \geq \gamma\kappa$. The claim follows by applying Lemma 2 on each point in $S_i \cap G$ as a ball, and each set $P_x$ as a bin with weight $|P_x|$. □

---
**Algorithm 2: Augmented-Summary-Outliers**$(X, k, t)$
---
**Input** : dataset $X$, number of centers $k$, number of outliers $t$
**Output**: a weighted dataset $Q$ as a summary of $X$

1 run *Summary-Outliers*$(X, k, t)$ (Algorithm 1) and obtain $X_r$ and $S = \cup_{i=0}^{r-1} S_i$
2 construct a set $S'$ of size $|X_r| - |S|$ by randomly sampling (with replacement) from $X \backslash (X_r \cup S)$
3 for each $x \in X \backslash X_r$, set $\pi(x) \leftarrow \arg\min_{y \in S \cup S'} d(x, y)$
4 for each $x \in X_r \cup (\cup_{i=0}^{r-1} S_i)$, assign weight $w_x \leftarrow |\pi^{-1}(x)|$ and add $(x, w_x)$ into $Q$
5 **return** $Q$
---

Lemma 5 follows directly from Lemma 6 and Lemma 7.

**The running time.** We now analyze the running time of Algorithm 1. At the $i$-th iteration, the sampling step at Line 6 can be done in $O(|X_i|)$ time. The nearest-center assignments at Line 7 and 9 can be done in $|S_i| \cdot |X_i| = O(\kappa |X_i|)$ time. Line 8 can be done by first sorting the distances in the increasing order and then scanning the shorted list until we get enough points. In this way the running time is bounded by $|X_i| \log |X_i| = O(\kappa |X_i|)$. Thus the total running time can be bounded by

$$\sum_{i=0,1,\ldots,r-1} O(\kappa |X_i|) = O(\kappa n) = O(\max\{\log n, k\} \cdot n),$$

where the first equation holds since the size of $X_i$ decreases geometrically, and the second equation is due to the definition of $\kappa$.

Finally, we comment that we can get a similar result for $(k, t)$-means by appropriately adjusting various constant parameters in the proof.

**Corollary 2** *Let $X_r$ and $\sigma : X \to X$ be computed by Algorithm 1. We have with probability* $(1 - 1/n^2)$ *that*

$$\sum_{x \in X \backslash X_r} d^2(x, \sigma(x)) = O\left(\mathsf{OPT}_{k,t}^{\mathsf{mea}}(X)\right).$$

We note that in the proof for the median objective function we make use of the triangle inequality in various places, while for the means objective function where the distances are squared, the triangle inequality does not hold. However we can instead use the inequality $2(x^2 + y^2) \geq (x + y)^2$, which will only make the constant parameters in the proofs slightly worse.

### 3.3 An Augmentation

In the case when $t \gg k$, which is typically the case in practice since the number of centers $k$ does not scale with the size of the dataset while the number of outliers $t$ does, we add an augmentation procedure to Algorithm 1 to achieve a better practical performance.

The procedure is presented in Algorithm 2.

The augmentation is as follows, after computing the set of outliers $X_r$ and the set of centers $S = \cup_{i=0}^{r-1} S_i$ in Algorithm 1, we sample randomly from $X \backslash (X_r \cup S)$ an additional set of center points $S'$ of size $|X_r| - |S|$. That is, we try to make the number of centers and the number of outliers in the summary to be balanced. We then reassign each point in the set $X \backslash X_r$ to its nearest center in $S \cup S'$. Denote the new mapping by $\pi$. Finally, we include points in $X_r$ and $S$, together with their weights, into the summary $Q$.

It is clear that the augmentation procedure preserves the size of the summary asymptotically. And by including more centers we have $\mathrm{loss}(Q) \leq \phi_X(\pi) \leq \phi_X(\sigma)$, where $\sigma$ is the mapping returned by Algorithm 1. The running time will increase to $O(tn)$ due to the reassignment step, but our algorithm is still much faster than all the baseline algorithms, as we shall see in Section 5.

## 4 Distributed Clustering with Outliers

In this section we discuss distributed $(k, t)$-median/means using the summary constructed in Algorithm 1. We will first discuss the case where the data is randomly partitioned among the $s$ sites, which

---
**Algorithm 3: Distributed-Median**$(A_1, \ldots, A_s, k, t)$
---
**Input** : For each $i \in [s]$, Site $i$ gets input dataset $A_i$ where $(A_1, \ldots, A_s)$ is a random partition of $X$

**Output** : a $(k, t)$-median clustering for $X = \cup_{i \in [s]} A_i$

1  for each $i \in [s]$, Site $i$ constructs a summary $Q_i$ by running *Summary-Outliers*$(A_i, k, 2t/s)$ (Algorithm 1) and sends $Q_i$ to the coordinator

2  the coordinator then performs a second level clustering on $Q = Q_1 \cup Q_2 \cup \ldots \cup Q_s$ using an off-the-shelf $(k, t)$-median algorithm, and returns the resulting clustering.
---

is the case in all of our experiments. The algorithm is presented in Algorithm 3. We will discuss the adversarial partition case at the end. We again only show the results for $(k, t)$-median since the same results will hold for $(k, t)$-means by slightly adjusting the constant parameters.

We will make use of the following known results. The first lemma says that the sum of costs of local optimal solutions that use the same number of outliers as the global optimal solution does is upper bounded by the cost of the global optimal solution.

**Lemma 8 (Guha *et al.* (2017))** *For each $i \in [s]$, let $t_i = |A_i \cap O^*|$ where $O^*$ is the set of outliers produced by the optimal $(k, t)$-median algorithm on $X = A_1 \cup A_2 \cup \ldots \cup A_s$. We have*

$$\sum_{i \in [s]} \mathsf{OPT}_{k,t_i}^{\mathsf{med}}(A_i) \leq O\left(\mathsf{OPT}_{k,t}^{\mathsf{med}}(X)\right).$$

The second lemma is a folklore for two-level clustering.

**Lemma 9 (Guha *et al.* (2003, 2017))** *Let $Q = Q_1 \cup Q_2 \cup \ldots \cup Q_s$ be the union of the summaries of the $s$ local datasets, and let $\mathsf{SOL}_{k,t}^{\mathsf{med}}(\cdot)$ be the cost function of a $\gamma$-approximation algorithm for $(k, t)$-median. We have*

$$\mathsf{SOL}_{k,t}^{\mathsf{med}}(Q) \leq O(\gamma) \cdot \left(\sum_{i \in [s]} loss(Q_i) + \mathsf{OPT}_{k,t}^{\mathsf{med}}(X)\right).$$

Now by Lemma 8, Lemma 9 and Theorem 1, we have that with probability $1 - 1/n$, $\mathsf{SOL}_{k,t}^{\mathsf{med}}(Q) \leq O(\gamma) \cdot \mathsf{OPT}_{k,t}^{\mathsf{med}}(X)$. And by Chernoff bounds and a union bound we have $t_i \leq 2t/s$ for all $i$ with probability $1 - 1/n^2$.[4]

**Theorem 2** *Suppose Algorithm 3 uses a $\gamma$-approximation algorithm for $(k, t)$-median in the second level clustering (Line 2). We have with probability $(1 - 1/n)$ that:*

- *it outputs a set of centers $C \subseteq \mathbb{R}^d$ and a set of outliers $O \subseteq X$ such that $\phi(X \backslash O, C) \leq O(\gamma) \cdot \mathsf{OPT}_{k,t}^{\mathsf{med}}(X)$;*

- *it uses one round of communication whose cost is bounded by $O(sk \log n + t)$;*

- *the running time at the $i$-th site is bounded by $O(\max\{\log n, k\} \cdot |A_i|)$, and the running time at the coordinator is that of the second level clustering.*

We note that in Mettu and Plaxton (2002) it was shown that under some mild assumption, $\Omega(kn)$ time is necessary for any $O(1)$-approximate randomized algorithm to compute $k$-median on $n$ points with nonnegligible success probability (e.g., $1/100$). Thus the running time of our algorithm is optimal up to a $\log n$ factor under the same assumption.

In the case that the dataset is adversarially partitioned, the total communication increases to $O(s(k \log n + t))$. This is because all of the $t$ outliers may go to the same site and thus $2t/s$ in Line 1 needs to be replaced by $t$.

Finally, we comment that the result above also holds for the summary constructed using the augmented version (Sec. 3.3), except, as discussed in Section 3, that the local running time at the $i$-th site will increase to $O(t|A_i|)$.

# 5 Experiments

## 5.1 Experimental Setup

### 5.1.1 Datasets and Algorithms

We make use of the following datasets.

- gauss-$\sigma$. This is a synthetic dataset, generated as follows: we first sample 100 centers from $[0, 1]^5$, i.e., each dimension is sampled uniformly at random from $[0, 1]$. For each center $c$, we generate 10000 points by adding each dimension of $c$ a random value sampled from the normal distribution $\mathcal{N}(0, \sigma)$. This way, we obtain $100 \cdot 10000 = 1$M points in total. We next construct the outliers as follows: we sample 5000 points from the 1M points, and for each sampled point, we add a random shift sampled from $[-2, 2]^5$.

- kddFull. This dataset is from 1999 kddcup competition and contains instances describing connections of sequences of tcp packets. There are about 4.9M data points. [5] We only consider the 34 numerical features of this dataset. We also normalize each feature so that it has zero mean and unit standard deviation. There are 23 classes in this dataset, 98.3% points of the dataset belong to 3 classes (normal 19.6%, neptune 21.6%, and smurf 56.8%). We consider small clusters as outliers and there are 45747 outliers.

- kddSp. This data set contains about 10% points of kddFull (released by the original provider). This dataset is also normalized and there are 8752 outliers.

- susy-$\Delta$. This data set has been produced using Monte Carlo simulations by Baldi *et al.* (2014). Each instance has 18 numerical features and there are 5M instances in total.[6]. We normalize each feature as we did in kddFull. We manually add outliers as follows: first we randomly sample 5000 data points; for each data point, we shift each of its dimension by a random value chosen from $[-\Delta, \Delta]$.

- Spatial-$\Delta$. This dataset is about 3D road network with elevation information from North Jutland, Denmark. It is designed for clustering and regression tasks. There are about 0.4M data points with 4 features. We normalize each feature so that it has zero mean and unit standard deviation. We add outliers as we did for susy-$\Delta$. [7]

Finding appropriate $k$ and $t$ values for the task of clustering with outliers is a separate problem, and is not part of the topic of this paper. In all our experiments, $k$ and $t$ are naturally suggested by the datasets we use unless they are unknown.

We compare the performance of following algorithms, each of which is implemented using the MPI framework and run in the coordinator model. The data are randomly partitioned among the sites.

- ball-grow. Algorithm 3 proposed in this paper, with the augmented version Algorithm 1 for the summary construction. As mentioned we use $k$-means-- as the second level clustering at Line 2. We fix $\alpha = 2$ and $\beta = 4.5$ in the subroutine Algorithm 1.

- rand. Each site constructs a summary by randomly sampling points from its local dataset. Each sampled point $p$ is assigned a weight equal to the number of points in the local dataset that are closer to $p$ than other points in the summary. The coordinator then collects all weighted samples from all sites and feeds to $k$-means-- for a second level clustering.

- $k$-means++. Each site constructs a summary of the local dataset using the $k$-means++ algorithm Arthur and Vassilvitskii (2007), and sends it to the coordinator. The coordinator feeds the unions all summaries to $k$-means-- for a second level clustering.

- $k$-means$\|$. An MPI implementation of the $k$-means$\|$ algorithm proposed by Bahmani *et al.* (2012) for distributed $k$-means clustering. To adapt their algorithm to solve the outlier version, we increase the parameter $k$ in the algorithm to $O(k+t)$, and then feed the outputted centers to $k$-means-- for a second level clustering.

| dataset | algo | summarySize | $\ell_1$-loss | $\ell_2$-loss | preRec | prec | recall |
|---|---|---|---|---|---|---|---|
| gauss-0.1 | ball-grow | 2.40e+4 | **2.08e+5** | **4.80e+4** | **0.9890** | **0.9951** | **0.9431** |
| | $k$-means++ | 2.40e+4 | 2.10e+5 | 5.50e+4 | 0.5740 | 0.9750 | 0.5735 |
| | $k$-means$\|$ | 2.50e+4 | 2.10e+5 | 5.40e+4 | 0.6239 | 0.9916 | 0.6235 |
| | rand | 2.04e+4 | 2.17e+5 | 6.84e+4 | 0.0249 | 0.2727 | 0.0249 |
| gauss-0.4 | ball-grow | 2.40e+4 | **4.91e+5** | **2.72e+5** | **0.8201** | 0.7915 | **0.7657** |
| | $k$-means++ | 2.40e+4 | 4.97e+5 | 2.82e+5 | 0.2161 | 0.6727 | 0.2091 |
| | $k$-means$\|$ | 2.50e+4 | 4.96e+5 | 2.79e+5 | 0.2573 | **0.7996** | 0.2458 |
| | rand | 2.40e+4 | 4.99e+5 | 2.90e+5 | 0.0234 | 0.2170 | 0.0212 |

Table 2: Clustering quality on gauss-$\sigma$ dataset, $k = 100$, $t = 5000$

### 5.1.2 Measurements

Let $C$ and $O$ be the sets of centers and outliers respectively returned by a tested algorithm. To evaluate the quality of the clustering results we use two metrics: (a) $\ell_1$-loss (for $(k, t)$-median): $\sum_{p \in X \setminus O} d(p, C)$; (b) $\ell_2$-loss (for $(k, t)$-means): $\sum_{p \in X \setminus O} d^2(p, C)$.

To measure the performance of outlier detection we use three metrics. Let $S$ be the set of points fed into the second level clustering $k$-means-- in each algorithm, and let $O^*$ be the set of actual outliers (i.e., the ground truth), we use the following metrics: (a) preRec: the proportion of actual outliers that are included in the returned summary, defined as $\frac{|S \cap O^*|}{|O^*|}$; (b) recall: the proportion of actual outliers that are returned by $k$-means--, defined as $\frac{|O \cap O^*|}{|O^*|}$; (c) prec: the proportion of points in $O$ that are actually outliers, defined as $\frac{|O \cap O^*|}{|O|}$.

### 5.1.3 Computation Environments

All algorithms are implemented in C++ with Boost.MPI support. We use Armadillo Sanderson (2010) as the numerical linear library and -O3 flag is enabled when compile the code. All experiments are conducted in a PowerEdge R730 server equipped with 2 x Intel Xeon E5-2667 v3 3.2GHz. This server has 8-core/16-thread per CPU, 192GB Memeory and 1.6TB SSD.

## 5.2 Experimental Results

We now present our experimental results. All results take the average of 10 runs.

### 5.2.1 Quality

We first compare the qualities of the summaries returned by ball-grow, rand and $k$-means$\|$. Note that the size of the summary returned by ball-grow is determined by the parameters $k$ and $t$, and we can not control the exact size. In $k$-means$\|$, the summary size is determined by the sample ratio, and again we can not control the exact size. On the other hand, the summary sizes of rand and $k$-means++ can be fully controlled. To be fair, we manually tune those parameters so that the sizes of summaries returned by different algorithms are roughly the same (the difference is less than 10%). In this set of experiments, each dataset is randomly partitioned into 20 sites.

Table 2 presents the experimental results on gauss datasets with different $\sigma$. We observe that ball-grow consistently gives better $\ell_1$-loss and $\ell_2$-loss than $k$-means$\|$ and $k$-means++, and rand performs the worst among all.

For outlier detection, rand fails completely. In both gauss-0.1 and gauss-0.4, ball-grow outperforms $k$-means++ and $k$-means$\|$ in almost all metrics. $k$-means$\|$ slightly outperforms $k$-means++. We also observe that in all gauss datasets, ball-grow gives very high preRec, i.e., the outliers are very likely to be included in the summary constructed by ball-grow.

Table 3 presents the experimental results on kddSp and kddFull datasets. In this set of experiments, ball-grow again outperforms its competitors in all metrics. Note that $k$-means$\|$ does not scale to kddFull.

| dataset | algo | summarySize | $\ell_1$-loss | $\ell_2$-loss | preRec | prec | recall |
|---|---|---|---|---|---|---|---|
| kddSp | ball-grow | 3.37e+4 | **8.00e+5** | **3.46e+6** | **0.6102** | **0.5586** | **0.5176** |
| | $k$-means++ | 3.37e+4 | 8.38e+5 | 4.95e+6 | 0.3660 | 0.3676 | 0.1787 |
| | $k$-means‖ | 3.30e+4 | 8.18e+5 | 4.19e+6 | 0.2921 | 0.3641 | 0.1552 |
| | rand | 3.37e+4 | 8.85e+5 | 1.06e+7 | 0.0698 | 0.5076 | 0.0374 |
| kddFull | ball-grow | 1.83e+5 | **7.38e+6** | **3.54e+7** | **0.7754** | **0.5992** | **0.5803** |
| | $k$-means++ | 1.83e+5 | 8.21e+6 | 4.65e+7 | 0.2188 | 0.2828 | 0.1439 |
| | $k$-means‖ | | | does not stop after 8 hours | | | |
| | rand | 1.83e+5 | 9.60e+6 | 1.11e+8 | 0.0378691 | 0.6115 | 0.0241 |

Table 3: Clustering quality. $k = 3$, $t = 8752$ for kddSp and $t = 45747$ for kddFull

| dataset | algo | summarySize | $\ell_1$-loss | $\ell_2$-loss | preRec | prec | recall |
|---|---|---|---|---|---|---|---|
| susy-5 | ball-grow | 2.40e+4 | **1.10e+7** | **2.76e+7** | **0.7508** | 0.6059 | **0.5933** |
| | $k$-means++ | 2.40e+4 | 1.11e+7 | 2.79e+7 | 0.1053 | 0.5678 | 0.1047 |
| | $k$-means‖ | 2.50e+4 | 1.11e+7 | 2.77e+7 | 0.1735 | **0.7877** | 0.1609 |
| | rand | 2.40e+4 | 1.12e+7 | 2.84e+7 | 0.004 | 0.2080 | 0.004 |
| susy-10 | ball-grow | 2.40e+4 | 1.11e+7 | **2.77e+7** | **0.9987** | 0.9558 | **0.9542** |
| | $k$-means++ | 2.40e+4 | 1.11e+7 | 2.90e+7 | 0.3412 | 0.8602 | 0.3412 |
| | $k$-means‖ | 2.49e+4 | 1.11e+7 | 2.84e+7 | 0.4832 | **0.9801** | 0.4823 |
| | rand | 2.40e+4 | 1.12e+7 | 3.08e+7 | 0.0047 | 0.2481 | 0.0047 |

Table 4: Clustering quality on susy dataset, $k = 100$, $t = 5000$

Table 4 presents the experimental results for susy-$\Delta$ dataset. We can observe that ball-grow produces slightly better results than $k$-means‖, $k$-means++ and rand in $\ell_1$-loss and $\ell_2$-loss. For outlier detection, ball-grow outperforms $k$-means++ and $k$-means‖ significantly in terms of preRec and recall, while $k$-means‖ gives slightly better prec. Table 5 presents the results for Spatial-15 dataset, and ball-grow again outperforms all other baseline algorithms in all metrics.

| dataset | algo | summarySize | $\ell_1$-loss | $\ell_2$-loss | preRec | prec | recall |
|---|---|---|---|---|---|---|---|
| Spatial-15 | ball-grow | 1.80e+3 | **5.21e+5** | **7.19e+5** | **0.9993** | **0.9993** | **0.9993** |
| | $k$-means++ | 1.80e+3 | 5.30e+5 | 7.79e+5 | 0.7698 | 0.9954 | 0.7697 |
| | $k$-means‖ | 1.80e+3 | 5.28e+5 | 7.38e+5 | 0.9198 | 0.9986 | 0.9198 |
| | rand | 1.80e+3 | 5.35e+5 | 1.03e+6 | 0.0047 | 0.2105 | 0.0047 |

Table 5: Clustering quality on Spatial dataset, $k = 5$, $t = 400$

### 5.2.2 Communication Costs

We next compare the communication cost of different algorithms. Figure 1a presents the experimental results. The communication cost is measured by the number of points exchanged between the coordinator and all sites. In this set of experiments we only change the number of partitions (i.e., # of sites $s$). The summaries returned by all algorithms have almost the same size.

We observe that the communication costs of ball-grow, $k$-means++ and rand are almost independent of the number of sites. Indeed, ball-grow, $k$-means++ and rand all run in one round and their communication cost is simply the size of the union of the $s$ summaries. $k$-means‖ incurs significantly more communication, and it grows almost linearly to the number of sites. This is because $k$-means‖ grows its summary in multiple rounds; in each round, the coordinator needs to collect messages from all sites and broadcasts the union of those messages. When there are 20 sites, $k$-means‖ incurs 20 times more communication cost than its competitors.

### 5.2.3 Running Time

We finally compare the running time of different algorithms. All experiments in this part are conducted on kddSp dataset since $k$-means‖ does not scale to kddFull; similar results can also be observed on other datasets. The running time we show is only the time used to construct the input (i.e., the union

Figure 1: experiments on `kddSp` dataset

of the $s$ summaries) for the second level clustering, and we do not include the running time of the second level clustering since it is always the same for all tested algorithms (i.e., the `k-means--`).

Figure 1b shows the running time when we change the number of sites while fix the size of the summary produced by each site. We observe that `k-means`‖ uses significantly more time than `ball-grow`, `k-means++` and `rand`. This is predictable because `k-means`‖ runs in multiple rounds and communicates more than its competitors. `ball-grow` uses significantly less time than others, typically $1/25$ of `k-means`‖, $1/7$ of `k-means++` and $1/2$ of `rand`. The reason that `ball-grow` is even faster than `rand` is that `ball-grow` only needs to compute weights for about half of the points in the constructed summary. As can be predicted, when we increase the number of sites, the total running time of each algorithm decreases.

We also investigate how the size of the summary will affect the running time. Note that for `ball-grow` the summary size is controlled by the parameter $t$. We fix $k = 3$ and vary $t$, resulting different summary sizes for `ball-grow`. For other algorithms, we tune the parameters so that they output summaries of similar sizes as `ball-grow` outputs. Figure 1c shows that when the size of summary increases, the running time increases almost linearly for all algorithms.

### 5.2.4 Stability of The Experimental Results

Our experiments involve some randomness and we have already averaged the experimental results for multiple runs to reduce the variance. To show that the experimental results are reasonably stable, we add Table 6 to present the standard deviations of the results. For each metric of a given algorithm, we gather $5$ data points, each of which is the averaged result of $10$ runs. We then calculate the mean/stddev of the $5$ data points.

| algo | $\ell_1$-loss | $\ell_2$-loss | preRec | prec | recall |
|---|---|---|---|---|---|
| ball-grow | $8.16\mathbf{E}5 \pm 1.1\mathbf{E}4$ | $3.46\mathbf{E}6 \pm 4.1\mathbf{E}5$ | $0.61 \pm 0.002$ | $0.55 \pm 0.007$ | $0.52 \pm 0.004$ |
| $k$-means++ | $8.83\mathbf{E}5 \pm 6.9\mathbf{E}4$ | $5.11\mathbf{E}6 \pm 2.8\mathbf{E}5$ | $0.37 \pm 0.004$ | $0.36 \pm 0.004$ | $0.18 \pm 0.002$ |
| $k$-means‖ | $8.41\mathbf{E}5 \pm 5.0\mathbf{E}4$ | $4.19\mathbf{E}6 \pm 1.4\mathbf{E}5$ | $0.29 \pm 0.004$ | $0.36 \pm 0.005$ | $0.16 \pm 0.004$ |
| rand | $9.20\mathbf{E}5 \pm 5.9\mathbf{E}4$ | $1.08\mathbf{E}7 \pm 2.0\mathbf{E}5$ | $0.07 \pm 0.001$ | $0.49 \pm 0.009$ | $0.04 \pm 0.005$ |

Table 6: Clustering quality on `kddSp` $k = 3$, $t = 8752$. Each entry is in the format of mean±stddev.

It can be seen from Table 6 that the results of our experiments are very stable in almost all metrics. $\ell_1$-`loss` is the only metric where our algorithm has some overlap with other baseline algorithms, but it is still safe to conclude that our algorithm outperforms all the baselines in almost all metrics. The similar stability is observed on other datasets.

### 5.2.5 Summary

We observe that `ball-grow` gives the best performance in almost all metrics for measuring summary quality. `k-means`‖ slightly outperforms `k-means++`. `rand` fails completely in the task of outliers detection. For communication, `ball-grow`, `k-means++` and `rand` incur similar costs and are independent of the number of sites. `k-means`‖ communicates significantly more than others. For running time, `ball-grow` runs much faster than others, while `k-means`‖ cannot scale to large-scale datasets.

**Acknowledgments**

Jiecao Chen, Erfan Sadeqi Azer and Qin Zhang are supported in part by NSF CCF-1525024, NSF CCF-1844234 and IIS-1633215.

## Footnotes

[1] We say an algorithm $\gamma$-approximates a problem if it outputs a solution that is at most $\gamma$ times the optimal solution.

[2] We say a solution is an $(a, b)$-approximation if the cost of the solution is $a \cdot C$ while excluding $b \cdot t$ points, where $C$ is the cost of the optimal solution excluding $t$ points.

[3] $\tilde{O}(\cdot)$ hides some logarithmic factors.

[4] For the convenience of the analysis we have assumed $t/s \geq \Omega(\log n)$, which is justifiable in practice since $t$ typically scales with the size of the dataset while $s$ is usually a fixed number.

[5]More information can be found in `http://kdd.ics.uci.edu/databases/kddcup99/kddcup99.html`

[6]More information about this dataset can be found in `https://archive.ics.uci.edu/ml/datasets/SUSY`

[7]More information can be found in `https://archive.ics.uci.edu/ml/datasets/3D+Road+Network+(North+Jutland,+Denmark)`.