[Reviews · NeurIPS 2018]

Reviewer 1



This paper investigates distributed K-means clustering with outlier detection, following a simultaneous communication setting in which data are partitioned across multiple sites. The method follows a standard two-level clustering. The first-level performs a summary of the local data set of each site. At the second level, a coordinator performs clustering using all the summaries. The main contribution of this work is in constructing simple summaries at the sites, which runs in time O(max{k; log n} n), with n the size of the data set and k the maximum number of cluster prototypes. The authors report experiments on both real and synthetic data, showing competitive performances of the proposed algorithm. The paper is clear and very well written. The experiments are comprehensive. The approach is simple, which is good. It is, therefore, fast. Several of the existing algorithms, which provide approximation guarantee in the same setting, require high-order polynomial time (quadratic or higher in n). This makes the proposed algorithm quite practical for large-scale data set. Having said that, I am not very familiar with the literature in these distributed and simultaneous communication settings. So, in summary, if there is no competing method in the literature with similar or lower complexity, then this is a nice work that tackles an important problem (definitely worth publication). Apart from this, I do not have any serious concern about the submission. Opinion after rebuttal: The author response is convincing to me and there is consensus among the reviewers that the paper is interesting. I am keeping my initial score.

Reviewer 2



The paper addresses the problem of performing the distributed k-mean/median clustering in the presence of outliers, and at the same time identifying the outliers. Data are partitioned across multiple sites either adversarially or randomly, and the sites and a central coordinator work jointly by communications to get the data clustering and the outliers. The authors proposed a practical algorithm with bounded running time $O(max{k,\log n} n)$, and bounded communication cost $O(s(k\log n+t))$ and $O(sk\log n+t)$ for adversarial and random data partitioning respectively, for a dataset with n data points, k centers, t outliers, and partitioned across s sites. They used a traditional two-level clustering framework (Guha et al. 2017). If using a $\gamma$-approximation algorithm for (k,t)-mean/median as the second-level clustering algorithm, their distributed algorithm has a bounded $O(\gamma)$ approximation factor. Extensive experimental studies were conducted to compare the performance of their algorithm with three baseline algorithms. The problem studied is very interesting and obviously has strong practical motivations. The authors can provide theoretical guarantees on the approximation factor, and upper bounds on the size of summary, the communication cost and the computation time. This is excitingly favourable. The analysis in bounding the approximation factor (Theorem 1) is interesting. The paper is pretty well written and is easy to follow. However, the summary construction algorithm and the distributed k-mean/median algorithm seem to largely follow Mettu and Plaxton 2002 and Guha et al. 2003, 2017. The paper does not provide significant algorithmic insights and a complete understanding, as existing clustering based NIPS papers, e.g. Chen et al. 2016. Currently the authors have upper bounds on the computation time and the communication cost. Is it possible to derive the time lower bound, as in Mettu and Plaxton 2002, or the communication lower bound, as in Chen et al. 2016? Only after deriving those lower bounds, they can have a complete understanding on the problem studied. Therefore, it might not be qualified as a seminar paper to get in NIPS. Furthermore, it lacks comprehensive experimental studies. The current experimental work focuses on two real-world datasets KddSp and KddFull. However, KddSp is only a subset of KddFull! More evaluation on real-world datasets are required before drawing the conclusion. The other two datasets in the supp. material are both synthetic datasets.

Reviewer 3



The paper considers the problem of computing a weighted summary for the k-means problem with outliers, in order to use this summary in a distributed setting. The algorithm is a variation of an approximation algorithm by Mettu and Plaxton (without outliers). The idea is: Iteratively, sample points, assign every point to its closest sample point, and then cut out a ball with the closest points around every sample point. The points that are "cut out" are then represented by their closest sample point and deleted. This is repeated until there are only very few points left. The adaptation considered in the paper for the outlier case is that the algorithm is stopped earlier, namely, when there are only 8*t points left (t being the number of outliers). These are then represented by themselves. The final coreset size is O(k log n + t), where the log n-term stems from the fact that in every round, log n points are sampled, and the t is from the last 8t points which are added to the summary. The algorithm is slightly improved for the case that t is much larger than k, which seems like a reasonable assumption. The paper experimentally compares the resulting algorithm to using k-means++ to construct a summary (k-means++ sampling, used to find O(k+t) samples, can also be used to get a O(1)-summary for k-means with outliers), and to kmeans||, again a k-means algorithm run with k'=O(k+t). The summaries are then clustered by using k-means-- to obtain a final solution (for all three approaches). Question: What does the sentence "ball-grow outperforms k-means++ and k-means|| in almost all metrics" mean? I do not find the entry where it does not outperform the others. My evaluation of this paper is two-fold. While in my opinion there are a couple of issues with the theoretical presentation of this work, the experiments look very good. I particularly like the evaluation with respect to actual outlier recognition. Still, for me there is a question mark on how this compares to adaptations of better k-means streaming algorithms (see below). Thus, I pick my initial vote to be weak accept. (+) The experimental results look great. The loss is smaller, but in particular, the outlier detection works much better. I like how the data set was constructed; this makes sense and gives a reasonable test case. I assume that "rand" was included to prove that there is need for a better method; rand is a really bad method for this and does indeed perform badly. But also k-means++ performs surprisingly bad. (-) The related work section does not cover works on coresets for k-means and streaming k-means algorithms, which would set this paper's contribution into perspective. (see below). (-) Comparison with coreset-based algorithms: A coreset for k-means is a weighted summary which approximates the cost of any possible solution within a factor of epsilon. It can thus be used to compute a PTAS (for constant k or d), and also to compute O(1)-approximations. For k-means it is possible to construct a coreset of size O(k*d*eps^(-4)), see [1], the size for k-median is even lower. And coresets have one additional benefit: Since the error can be made arbitrarily small, it is possible to embedd them into a merge-and-reduce framework, which transforms them into streaming algorithms. These are then very fast and do not need random access any more! The construction in this paper, however, needs random access. While uniform sampling is easy and requires no extra computation (compared to, e.g., the sampling in k-means++, which needs a lot of precomputation), it needs time and in particular random access to cut out the balls around all sampled points. And, since the guarantee here is only O(1), merge and reduce will not compute a very good summary. (-) Comparison to streaming k-means algorithms: There are two types of these algorithms. First, there are coreset-inspired streaming algorithms. These are not mentioned in the paper at all. Second, there are streaming algorithms that are inspired by online facility location algorithms. Most relevant here is probably [2]. These methods are typically faster and need a size around O(k log n), but only provide a O(1) guarantee, like this paper. So for me, a natural question would be whether those methods could be extended to outliers. Again, the advantage of that would be that there is no need for random access. (-) It would be nice to have some justification for the choice of k-means--, which, as far as I can see, has no guarantee at all. Since there is the paper [3] which apparently implemented a constant factor approximation for k-means with outliers, one wonders if that would give more stable / informative information. I gather that k-means-- is probably a fair bit faster, yet it is unclear to me how high the variance of this algorithm is. For standard k-means, classical Lloyd is much more unreliable than k-means++, and I believe that k-means-- is simply a Lloyd variant for k-means with outliers? [1] Feldman, Langberg: A Unified Framework for Approximating and Clustering Data, STOC 2011, https://arxiv.org/abs/1106.1379 [2] Shindler, Wong, Meyerson: Fast and Accurate k-means For Large Datasets, NIPS 2011, www-bcf.usc.edu/~shindler/papers/FastKMeans_nips11.pdf. [3] Gupta, Kumar, Lu, Moseley, Vassilvitskii: Local Search Methods for k-Means with Outliers, PVLDB 2017. Edit after author feedback: The response is convincing to me. I still think that comparing to (non-outlier) work on coresets and streaming sets this paper into perspective, but I understand the points that are risen in the feedback. I thus increase my score to 7.